# Burden of post-COVID-19 syndrome and implications for healthcare service planning: A population-based cohort study

**Dominik Menges** [1], **Tala Ballouz** [1], **Alexia Anagnostopoulos** [1], **Hélène E. Aschmann** [1], **Anja Domenghino** [1,2], **Jan S. Fehr** [1], **Milo A. Puhan** [1]*

1 Epidemiology, Biostatistics and Prevention Institute (EBPI), University of Zurich (UZH), Zurich, Switzerland,
2 Department of Visceral and Transplantation Surgery, University Hospital Zurich, University of Zurich (UZH), Zurich, Switzerland

☯ These authors contributed equally to this work.
* miloalan.puhan@uzh.ch

## Abstract

### Background

Longer-term consequences after SARS-CoV-2 infection are becoming an important burden to societies and healthcare systems. Data on post-COVID-19 syndrome in the general population are required for the timely planning of healthcare services and resources. The objective of this study was to assess the prevalence of impaired health status and physical and mental health symptoms among individuals at least six months after SARS-CoV-2 infection, and to characterize their healthcare utilization.

### Methods

This population-based prospective cohort study (Zurich SARS-CoV-2 Cohort) enrolled 431 adults from the general population with polymerase chain reaction-confirmed SARS-CoV-2 infection reported to health authorities between 27 February 2020 and 05 August 2020 in the Canton of Zurich, Switzerland. We evaluated the proportion of individuals reporting not to have fully recovered since SARS-CoV-2 infection, and the proportion reporting fatigue (Fatigue Assessment Scale), dyspnea (mMRC dyspnea scale) or depression (DASS-21) at six to eight months after diagnosis. Furthermore, the proportion of individuals with at least one healthcare contact after their acute illness was evaluated. Multivariable logistic regression models were used to assess factors associated with these main outcomes.

### Results

Symptoms were present in 385 (89%) participants at diagnosis and 81 (19%) were initially hospitalized. At six to eight months, 111 (26%) reported not having fully recovered. 233 (55%) participants reported symptoms of fatigue, 96 (25%) had at least grade 1 dyspnea, and 111 (26%) had DASS-21 scores indicating symptoms of depression. 170 (40%) participants reported at least one general practitioner visit related to COVID-19 after acute illness, and 10% (8/81) of initially hospitalized individuals were rehospitalized. Individuals that have

stratification variables used for sensitivity analysis cannot be shared publicly to ensure anonymity of study participants.

**Funding:** The Zurich SARS-CoV-2 Cohort study is part of the Corona Immunitas research program, coordinated by the Swiss School of Public Health (SSPH+) and funded through SSPH+ fundraising, including funding by the Swiss Federal Office of Public Health, the Cantons of Switzerland (Basel, Vaud and Zurich), private funders (ethical guidelines for funding stated by SSPH+ were respected) and institutional funds of the participating universities. Additional funding specific to this study was provided by the Department of Health of the Canton of Zurich and the University of Zurich Foundation. Study funders had no role in study design, data collection and analysis, interpretation, decision to publish or preparation of this manuscript.

**Competing interests:** The authors have declared that no competing interests exist.

not fully recovered or suffer from fatigue, dyspnea or depression were more likely to have further healthcare contacts. However, a third of individuals (37/111) that have not fully recovered did not seek further care.

## Conclusions

In this population-based study, a relevant proportion of participants suffered from longer-term consequences after SARS-CoV-2 infection. With millions infected across the world, our findings emphasize the need for the timely planning of resources and patient-centered services for post-COVID-19 care.

## Background

As of February 2021, the severe acute respiratory syndrome coronavirus 2 (SARS-CoV-2) pandemic has resulted in more than 110 million infected cases and almost 2.5 million lives lost, at significant costs to healthcare systems and societies worldwide [1]. While initial public health responses focused on reducing the acute burden of coronavirus disease 2019 (COVID-19), a growing body of evidence indicates that SARS-CoV-2 infection can also result in longer-term physical and mental health consequences, which are of increasing concern for healthcare systems [2–4]. Such consequences lasting for longer than three months after infection are currently referred to as "post-COVID-19 syndrome" or "Long Covid" [4].

Few observational studies, predominantly conducted in patients hospitalized for acute COVID-19, have examined the persistence of symptoms and development of complications after SARS-CoV-2 infection beyond three months [5–9]. These studies reported that 15% up to 76% of infected individuals may experience persistent complaints for at least six months after acute illness [5, 9]. Further studies in hospitalized patients found that up to 20% of patients had to be rehospitalized [10], and up to 80% may require follow-up in primary care within 2 months of hospital discharge [11]. However, current evidence shows that post-COVID-19 syndrome does not only occur in individuals with severe disease requiring hospitalization or in older individuals with comorbidities, but also in young and previously healthy individuals with mild disease [3, 7, 9, 12, 13]. Data regarding the full burden of post-COVID-19 syndrome in the broader population of infected individuals is currently lacking. It is increasingly acknowledged that specific healthcare services and resources will be required to support the needs of individuals suffering from post-COVID-19 syndrome [2, 4, 14]. In response, several countries have started to set up specialized clinics [15, 16], and multiple patient support groups and networks for affected individuals have been formed to improve the general understanding of post-COVID-19 syndrome and identify needs for healthcare systems [14].

To successfully plan healthcare services and efficiently allocate public health resources, it is essential to determine the burden of longer-term consequences of SARS-CoV-2 infection and the needs of affected individuals. In this population-based study of SARS-CoV-2 infected individuals surveyed at least six months after diagnosis, we aimed to assess the longer-term physical and mental health impact of COVID-19 and the associated healthcare utilization. Thereby, we aimed to provide an evidence base for the planning of healthcare services for individuals suffering from post-COVID-19 syndrome.

## Methods

### Study design and participants

This study is based on data from participants of the Zurich SARS-CoV-2 Cohort study, a prospective, longitudinal cohort of polymerase chain reaction (PCR)-confirmed SARS-CoV-2 infected individuals diagnosed between 27 February 2020 and 05 August 2020. We recruited study participants from within contact tracing at the Department of Health of the Canton of Zurich, Switzerland, based on mandatory laboratory reporting of all individuals diagnosed with SARS-CoV-2. We screened all SARS-CoV-2-positive individuals for whom contact information was available for eligibility. Eligibility criteria were being aged 18 years or older and able to follow study procedures, having sufficient knowledge of the German language, and residing in the Canton of Zurich. We enrolled participants into the study between 06 October 2020 and 26 January 2021, at a median of 7.2 months (range 5.9 to 10.3 months) after their diagnosis. The study was prospectively registered on the International Standard Randomised Controlled Trial Number registry (ISRCTN14990068) and was approved by the responsible ethics committee of the Canton of Zurich, Switzerland (Kantonale Ethik-Kommission Zürich; BASEC-Nr. 2020–01739). Electronic or written informed consent was obtained from all participants.

### Data sources and outcome measurement

After enrolment, participants completed an electronic baseline questionnaire including questions on socio-demographics, medical comorbidities and risk factors, details on their acute SARS-CoV-2 infection, current health status and symptoms, healthcare contacts since diagnosis, and health-related quality of life. All data was collected through the Research Electronic Data Capture (REDCap) survey system.

Acute COVID-19 was defined as symptoms, consequences and healthcare contacts within four weeks of diagnosis. To capture the longer-term effects of SARS-CoV-2 infection, we evaluated whether participants who were symptomatic in the acute phase had fully recovered compared to their normal health status before infection using a four-category scale (i.e., feeling "recovered and symptom free", "better but not fully recovered", "neither better nor worse", or "worse"). We assessed the presence and type of any new or ongoing symptoms since the acute illness using a comprehensive list of symptoms. Additionally, we reviewed and coded comments in free text fields for further new or ongoing symptoms not captured by the preconceived questionnaire.

We evaluated the presence of fatigue using the Fatigue Assessment Scale (FAS), using a score of 22 or more as a threshold for determining the presence of relevant fatigue [17]. To assess longer-term respiratory complications, we administered the modified Medical Research Council (mMRC) dyspnea scale [18]. We assessed the presence and severity of depression, anxiety and stress symptoms using the 21-item Depression, Anxiety and Stress Scale (DASS-21). We calculated category scores as the sum of subscale item scores multiplied by two and assigned corresponding severity levels according to official user guidance [19, 20]. We evaluated health-related quality of life using the EQ-5D-5L instrument and visual analogue scale (EQ VAS) [21, 22]. We used the Dutch value set for calculating EQ-5D-5L index scores, as no value set or guidance on the most appropriate value set for Switzerland is available, and we judged the population of the Netherlands to be relatively similar to the Swiss population.

We assessed healthcare service utilization by eliciting all healthcare contacts that participants have had since their acute illness. We asked participants about any general practitioner visits, medical hotline calls, and hospital admissions, as well as the main reason for each

contact. To evaluate healthcare utilization specifically due to COVID-19, we restricted our analysis to healthcare contacts reported to be related to persistent or worsening symptoms, complications or new medical diagnoses related to COVID-19, or routine follow-up after COVID-19. In addition, we asked participants to report any medical conditions that have been newly diagnosed since their acute illness and whether the condition was evaluated as COVID-19-related by their physician or themselves.

## Statistical analysis

We used descriptive statistics to analyze participant characteristics and outcomes of interest, and present results for the entire study population as well as stratified by age groups, sex, and hospitalization status.

We examined the data for missing values and report such where applicable. For responses to the FAS and DASS-21 instruments, we replaced missing data with the mean of the scores from the other available items. This was done for a maximum of two missing values for the FAS and for a maximum of one value for the DASS-21. We omitted FAS or DASS-21 responses with higher amounts of missing data from respective analyses. For the EQ-5D-5L, responses with invalid and missing data were omitted given the small amount of missing data (n = 4). No imputation was applied for other missing data.

We assessed associations of potential predictors with the primary outcomes using univariable and multivariable logistic regression models. We based model selection on clinical and epidemiological reasoning and the Akaike Information Criterion (AIC). We defined age group, sex and initial hospitalization as a priori covariables in the models based on the findings of other studies. We performed model selection separately for each outcome of interest by including variables that improved model fit based on AIC, with a difference of 2 points considered relevant. For the outcome of (non-)recovery, we restricted the analysis to initially symptomatic participants since the respective question was conditional on the presence of symptoms at time of infection. We report regression analysis results as odds ratios (OR) with corresponding 95% confidence interval (CI) and two-sided Wald-type statistical test. No p-value adjustment was applied. All analyses were performed using R version 4.0.2 [23].

In sensitivity analyses, we stratified results into time periods with limited testing (period before 25 June 2020, testing restricted to high-risk or severely symptomatic individuals) and increased testing (period after 25 June 2020, all symptomatic individuals could be tested). Additionally, we stratified results into time periods with limited and high public awareness of post-COVID-19 syndrome (questionnaire completion in periods before and after 09 November 2020, when major Swiss news outlets started reporting about post-COVID-19 syndrome). Last, we descriptively compared participants and nonparticipants for age, sex, presence of symptoms and hospitalization at infection to assess potential selection bias.

## Results

### Study population

Between 27 February 2020 and 05 August 2020, 4639 individuals were diagnosed with SARS-CoV-2 in the Canton of Zurich (Fig 1). Contact information was available for 2209 individuals, among which 1309 were eligible and invited to participate in our study. 442 individuals agreed to participate (participation rate 34%) and 431 were included in this analysis.

The median age of participants was 47 years (IQR 33 to 58 years) and 50% were female (Table 1). At least one chronic comorbidity was reported by 147 (34%) participants. During acute infection, 385 (89%) participants were symptomatic, with a median of 6 (IQR 3 to 8) symptoms reported. Symptoms were described as mild to moderate in 221 (51%) and severe to

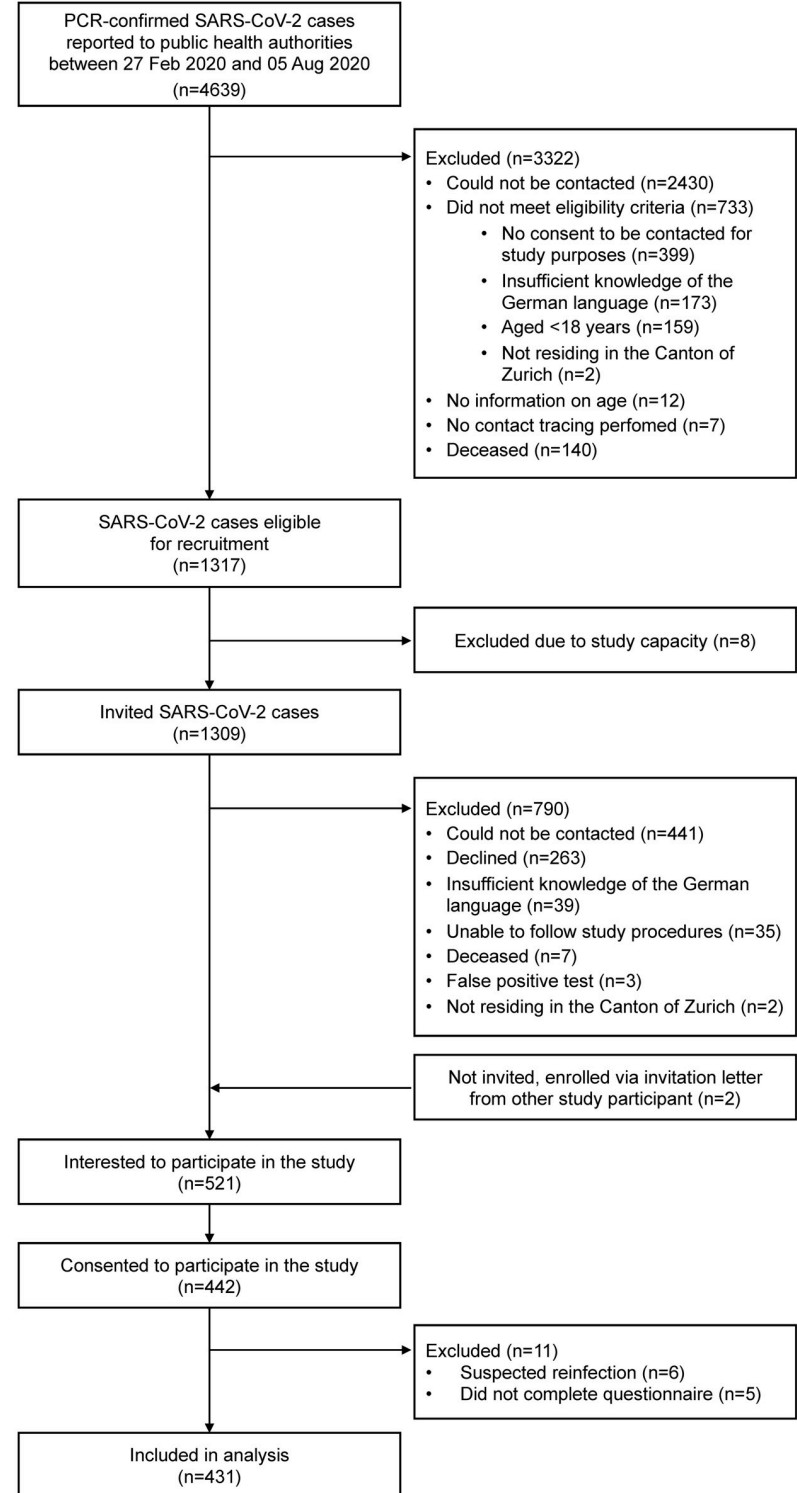

**Fig 1. Flow chart for the inclusion of SARS-CoV-2 infected individuals from the Canton of Zurich, diagnosed between 27 February 2020 and 05 August 2020.**

**Table 1. Characteristics of study participants enrolled in the Zurich SARS-CoV-2 cohort study.**

| Variable | N = 431 |
| --- | --- |
| **Age (years)** | |
| Median (IQR) | 47 (33 to 58) |
| **Age group (years)** | |
| 18–39 | 164 (38.1%) |
| 40–64 | 205 (47.6%) |
| ≥65 | 62 (14.4%) |
| **Sex** | |
| Female | 214 (49.7%) |
| Male | 217 (50.3%) |
| **Time since diagnosis (days)** | |
| Median (IQR) | 220 (181 to 232) |
| **Initial symptom severity** | |
| Asymptomatic | 46 (10.7%) |
| Mild to moderate | 221 (51.3%) |
| Severe to very severe | 164 (38.1%) |
| **Initial symptom count** | |
| Median (IQR) | 6 (3 to 8) |
| *Missing* | *1* |
| **Initial symptom duration (days)** | |
| Median (IQR) | 10 (6 to 20) |
| *Missing* | *8* |
| **Hospitalization and ICU stay** | |
| Non-hospitalized | 350 (81.2%) |
| Hospitalized without ICU stay | 71 (16.5%) |
| Hospitalized with ICU stay | 10 (2.3%) |
| Intubation during ICU stay (N = 10) | 7 (70%) |
| **Smoking status** | |
| Non-smoker | 245 (57.2%) |
| Ex-smoker | 122 (28.5%) |
| Smoker | 61 (14.3%) |
| *Missing* | *3* |
| **Body mass index (kg/m$^2$)** | |
| Median (IQR) | 24.8 (22.2 to 27.5) |
| *Missing* | *8* |
| **Comorbidities** | |
| No comorbidity | 283 (65.7%) |
| At least one comorbidity | 147 (34.1%) |
| *Missing* | *1* |
| **Education** | |
| None or mandatory school | 20 (4.7%) |
| Vocational training or specialized baccalaureate | 186 (43.5%) |
| Higher technical school or college | 106 (24.8%) |
| University | 116 (27.1%) |
| *Missing* | *3* |
| **Employment** | |
| Employed | 321 (75.4%) |
| Student | 14 (3.3%) |

(*Continued*)

**Table 1.** (Continued)

| Variable | N = 431 |
|---|---|
| Retired | 62 (14.6%) |
| Unemployed or other | 29 (6.8%) |
| *Missing* | *5* |
| **Income (CHF)** | |
| <6'000 | 133 (32.8%) |
| 6'000–12'000 | 156 (38.4%) |
| >12'000 | 117 (28.8%) |
| *Missing* | *25* |

CHF = Swiss Francs, ICU = Intensive Care Unit, IQR = Interquartile Range.

very severe in 164 participants (38%). Most commonly reported symptoms were fatigue (64%), fever (63%), cough (50%) and loss of taste or smell (49%). 81 (19%) participants were hospitalized due to COVID-19 for a median duration of 7 days (IQR 4 to 15 days).

Compared to individuals not participating in our study, participants in our study were younger on average, and a lower proportion was hospitalized for COVID-19 (19% compared to 24% of nonparticipants) (S1 Table).

## Recovery and longer-term symptoms

Overall, 111 (26%) participants reported that they had not fully recovered at six to eight months after SARS-CoV-2 infection (Table 2). A higher percentage of female participants and initially hospitalized individuals reported not having fully recovered compared to males and non-hospitalized individuals, respectively. In multivariable analyses among initially symptomatic participants, we found evidence that severe to very severe symptoms during acute illness and the presence of comorbidities were associated with non-recovery (Fig 2 and S2 Table). Furthermore, females were less likely to have recovered compared to males, while there was no evidence for an association of age or initial hospitalization with non-recovery.

Similarly, 106 (25%) participants reported new or ongoing symptoms, with a higher percentage in females compared to males (Table 2). Fatigue (12%), cough (10%), sore throat (9%) and headache (9%) were the most frequently reported symptoms. Taste and smell disturbances were reported by 21 (5%) and rash by 3 (1%) individuals. Among non-recovered participants, 51 (46%) also described experiencing new or ongoing symptoms. Meanwhile, 55 (52%) participants reporting such symptoms stated not having fully recovered. Overall, 166 (39%) participants reported either not having fully recovered or having new or ongoing symptoms.

## Fatigue

Among all participants, 233 (55%) participants having a score indicating fatigue, with a median FAS score of 22 (IQR 19 to 25) (Table 2). Younger individuals and female participants more frequently reported symptoms of fatigue compared to older age groups and males, respectively. In multivariable analyses, we found evidence that individuals aged 40 years or older were less likely to experience fatigue compared to 18–39 year-old participants (Fig 2 and S3 Table). However, we found no evidence for an association of sex, initial symptom severity or hospitalization with fatigue.

**Table 2. Relative health status, fatigue, dyspnea, mental health, and health-related quality of life in study participants at six to eight months after SARS-CoV-2 infection.**

| Variable | Age group | | | Sex | | Hospitalization | | Overall, N = 431 |
|---|---|---|---|---|---|---|---|---|
| | 18–39 years, N = 164 | 40–64 years, N = 205 | ≥65 years, N = 62 | Female, N = 214 | Male, N = 217 | Non-hospitalized, N = 350 | Hospitalized, N = 81 | |
| **Recovery** | | | | | | | | |
| Recovered to normal health status | 133 (81.1%) | 140 (68.3%) | 47 (75.8%) | 148 (69.2%) | 172 (79.3%) | 268 (76.6%) | 52 (64.2%) | 320 (74.2%) |
| Not recovered to normal health status | 31 (18.9%) | 65 (31.7%) | 15 (24.2%) | 66 (30.8%) | 45 (20.7%) | 82 (23.4%) | 29 (35.8%) | 111 (25.8%) |
| **Self-reported symptoms** | | | | | | | | |
| No new or ongoing symptoms | 120 (73.2%) | 151 (73.7%) | 54 (87.1%) | 151 (70.6%) | 174 (80.2%) | 263 (75.1%) | 62 (76.5%) | 325 (75.4%) |
| Any new or ongoing symptoms | 44 (26.8%) | 54 (26.3%) | 8 (12.9%) | 63 (29.4%) | 43 (19.8%) | 87 (24.9%) | 19 (23.5%) | 106 (24.6%) |
| **Recovery and symptoms** | | | | | | | | |
| Recovered and symptom-free | 105 (64.0%) | 116 (56.6%) | 44 (71.0%) | 114 (53.3%) | 151 (69.6%) | 221 (63.1%) | 44 (54.3%) | 265 (61.5%) |
| Not recovered or experiencing symptoms | 59 (36.0%) | 89 (43.4%) | 18 (29.0%) | 100 (46.7%) | 66 (30.4%) | 129 (36.9%) | 37 (45.7%) | 166 (38.5%) |
| **Fatigue (measured by FAS)** | | | | | | | | |
| No fatigue | 59 (36.0%) | 100 (49.0%) | 34 (58.6%) | 86 (40.8%) | 107 (49.8%) | 154 (44.1%) | 39 (50.6%) | 193 (45.3%) |
| Fatigue | 105 (64.0%) | 104 (51.0%) | 24 (41.4%) | 125 (59.2%) | 108 (50.2%) | 195 (55.9%) | 38 (49.4%) | 233 (54.7%) |
| *Missing* | *0* | *1* | *4* | *3* | *2* | *1* | *4* | *5* |
| **Dyspnea (measured by mMRC scale)** | | | | | | | | |
| mMRC grade 0 | 126 (82.4%) | 139 (74.3%) | 34 (61.8%) | 139 (70.9%) | 160 (80.4%) | 259 (81.4%) | 40 (51.9%) | 299 (75.7%) |
| mMRC grade 1 | 25 (16.3%) | 39 (20.9%) | 17 (30.9%) | 48 (24.5%) | 33 (16.6%) | 53 (16.7%) | 28 (36.4%) | 81 (20.5%) |
| mMRC grade ≥2 | 2 (1.3%) | 9 (4.8%) | 4 (7.3%) | 9 (4.6%) | 6 (3.0%) | 6 (1.9%) | 9 (11.7%) | 15 (3.8%) |
| *Missing* | *11* | *18* | *7* | *18* | *18* | *32* | *4* | *36* |
| **Depression (measured by DASS-21)** | | | | | | | | |
| No depression | 123 (75.0%) | 151 (74.0%) | 43 (71.7%) | 149 (70.6%) | 168 (77.4%) | 263 (75.4%) | 54 (68.4%) | 317 (74.1%) |
| Mild to moderate depression | 33 (20.1%) | 39 (19.1%) | 13 (21.7%) | 50 (23.7%) | 35 (16.1%) | 68 (19.5%) | 17 (21.5%) | 85 (19.9%) |
| Severe to very severe depression | 8 (4.9%) | 14 (6.9%) | 4 (6.7%) | 12 (5.7%) | 14 (6.5%) | 18 (5.2%) | 8 (10.1%) | 26 (6.1%) |
| *Missing* | *0* | *1* | *2* | *3* | *0* | *1* | *2* | *3* |
| **Anxiety (measured by DASS-21)** | | | | | | | | |
| No anxiety | 114 (69.5%) | 136 (67.3%) | 41 (68.3%) | 125 (59.5%) | 166 (76.9%) | 246 (70.7%) | 45 (57.7%) | 291 (68.3%) |
| Mild to moderate anxiety | 38 (23.2%) | 47 (23.3%) | 18 (30.0%) | 64 (30.5%) | 39 (18.1%) | 82 (23.6%) | 21 (26.9%) | 103 (24.2%) |
| Severe to very severe anxiety | 12 (7.3%) | 19 (9.4%) | 1 (1.7%) | 21 (10.0%) | 11 (5.1%) | 20 (5.7%) | 12 (15.4%) | 32 (7.5%) |
| *Missing* | *0* | *3* | *2* | *4* | *1* | *2* | *3* | *5* |
| **Stress (measured by DASS-21)** | | | | | | | | |
| No stress | 134 (82.2%) | 169 (82.8%) | 54 (93.1%) | 169 (80.1%) | 188 (87.9%) | 293 (84.2%) | 64 (83.1%) | 357 (84.0%) |
| Mild to moderate stress | 21 (12.9%) | 26 (12.7%) | 4 (6.9%) | 33 (15.6%) | 18 (8.4%) | 43 (12.4%) | 8 (10.4%) | 51 (12.0%) |
| Severe to very severe stress | 8 (4.9%) | 9 (4.4%) | 0 (0.0%) | 9 (4.3%) | 8 (3.7%) | 12 (3.4%) | 5 (6.5%) | 17 (4.0%) |
| *Missing* | *1* | *1* | *4* | *3* | *3* | *2* | *4* | *6* |
| **EQ-5D mobility** | | | | | | | | |
| No mobility problems | 156 (95.1%) | 180 (87.8%) | 45 (75.0%) | 188 (88.3%) | 193 (89.4%) | 324 (92.8%) | 57 (71.2%) | 381 (88.8%) |
| Mobility problems | 8 (4.9%) | 25 (12.2%) | 15 (25.0%) | 25 (11.7%) | 23 (10.6%) | 25 (7.2%) | 23 (28.7%) | 48 (11.2%) |
| *Missing* | *0* | *0* | *2* | *1* | *1* | *1* | *1* | *2* |
| **EQ-5D self care** | | | | | | | | |

(*Continued*)

**Table 2.** (Continued)

| Variable | Age group | | | Sex | | Hospitalization | | Overall, N = 431 |
|---|---|---|---|---|---|---|---|---|
| | 18–39 years, N = 164 | 40–64 years, N = 205 | ≥65 years, N = 62 | Female, N = 214 | Male, N = 217 | Non-hospitalized, N = 350 | Hospitalized, N = 81 | |
| No problems with self-care | 164 (100.0%) | 203 (99.0%) | 61 (100.0%) | 212 (99.5%) | 216 (99.5%) | 348 (99.4%) | 80 (100.0%) | 428 (99.5%) |
| Problems with self-care | 0 (0.0%) | 2 (1.0%) | 0 (0.0%) | 1 (0.5%) | 1 (0.5%) | 2 (0.6%) | 0 (0.0%) | 2 (0.5%) |
| *Missing* | *0* | *0* | *1* | *1* | *0* | *0* | *1* | *1* |
| **EQ-5D usual activities** | | | | | | | | |
| No problems during usual activities | 146 (89.0%) | 184 (89.8%) | 55 (90.2%) | 184 (86.4%) | 201 (92.6%) | 322 (92.0%) | 63 (78.8%) | 385 (89.5%) |
| Problems during usual activities | 18 (11.0%) | 21 (10.2%) | 6 (9.8%) | 29 (13.6%) | 16 (7.4%) | 28 (8.0%) | 17 (21.2%) | 45 (10.5%) |
| *Missing* | *0* | *0* | *1* | *1* | *0* | *0* | *1* | *1* |
| **EQ-5D pain & discomfort** | | | | | | | | |
| No pain or discomfort present | 124 (75.6%) | 124 (61.1%) | 29 (47.5%) | 132 (62.3%) | 145 (67.1%) | 241 (69.3%) | 36 (45.0%) | 277 (64.7%) |
| Pain or discomfort present | 40 (24.4%) | 79 (38.9%) | 32 (52.5%) | 80 (37.7%) | 71 (32.9%) | 107 (30.7%) | 44 (55.0%) | 151 (35.3%) |
| *Missing* | *0* | *2* | *1* | *2* | *1* | *2* | *1* | *3* |
| **EQ-5D anxiety & depression** | | | | | | | | |
| No anxiety or depression present | 107 (65.2%) | 144 (70.2%) | 46 (75.4%) | 131 (61.5%) | 166 (76.5%) | 241 (68.9%) | 56 (70.0%) | 297 (69.1%) |
| Anxiety or depression present | 57 (34.8%) | 61 (29.8%) | 15 (24.6%) | 82 (38.5%) | 51 (23.5%) | 109 (31.1%) | 24 (30.0%) | 133 (30.9%) |
| *Missing* | *0* | *0* | *1* | *1* | *0* | *0* | *1* | *1* |
| **EQ-5D-5L index score** | | | | | | | | |
| Median (IQR) | 1.00 (0.86 to 1.00) | 0.89 (0.85 to 1.00) | 0.89 (0.82 to 1.00) | 0.89 (0.82 to 1.00) | 1.00 (0.87 to 1.00) | 1.00 (0.86 to 1.00) | 0.88 (0.82 to 1.00) | 0.89 (0.85 to 1.00) |
| Range | 0.41 to 1.00 | 0.07 to 1.00 | 0.47 to 1.00 | 0.37 to 1.00 | 0.07 to 1.00 | 0.07 to 1.00 | 0.37 to 1.00 | 0.07 to 1.00 |
| *Missing* | *0* | *2* | *2* | *2* | *2* | *3* | *1* | *4* |
| **EQ VAS** | | | | | | | | |
| Median (IQR) | 85 (80 to 90) | 85 (77 to 90) | 80 (70 to 88) | 85 (77 to 90) | 85 (79 to 90) | 85 (80 to 90) | 80 (70 to 89) | 85 (77 to 90) |
| Range | 20 to 100 | 25 to 100 | 24 to 95 | 20 to 100 | 24 to 100 | 25 to 100 | 20 to 97 | 20 to 100 |
| *Missing* | *2* | *6* | *2* | *6* | *4* | *5* | *5* | *10* |

DASS-21 = Depression, Anxiety and Stress Score (21 items), EQ = EuroQol, FAS = Fatigue Assessment Scale, IQR = Interquartile Range, mMRC = modified Medical Research Council, VAS = Visual Analogue Scale.

## Dyspnea

A total of 96 (25%) participants reported to suffer from mMRC grade 1 dyspnea or higher (Table 2). We observed a higher percentage of grade ≥1 dyspnea among older age individuals, females, and initially hospitalized participants. In multivariable analyses, we found evidence for an association of grade ≥1 dyspnea with female sex, initial hospitalization, higher body mass index and presence of comorbidities, but not for initial symptom severity, smoking status or presence of a chronic respiratory condition (Fig 2 and S4 Table).

## Depression, anxiety and stress

Overall, 111 (26%) participants reported symptoms of depression, 135 (32%) reported symptoms of anxiety, and 68 (16%) reported symptoms of stress (Table 2). Higher proportions of participants reported depressive symptoms in older age groups and among females. Similar

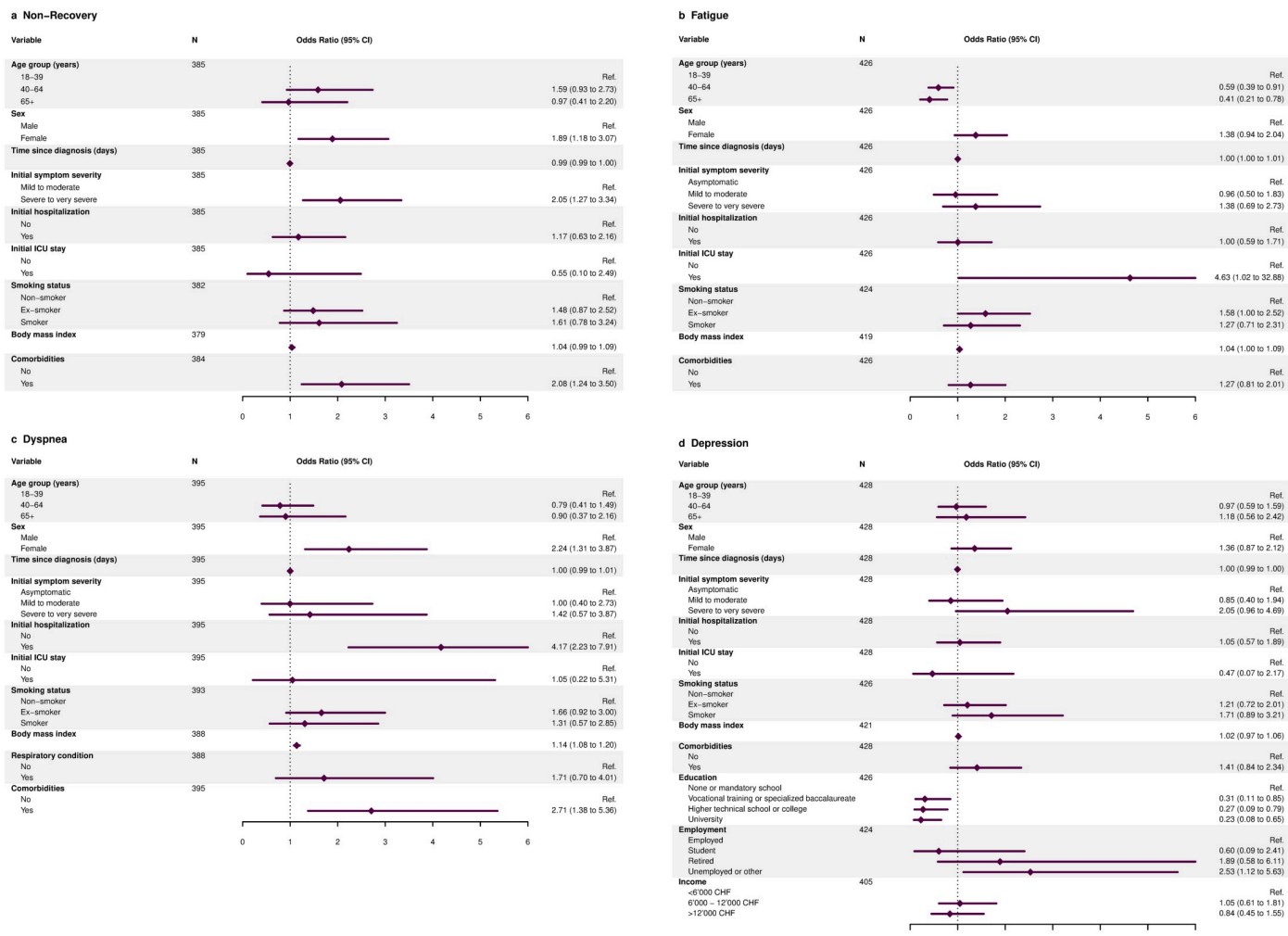

**Fig 2. Associations for non-recovery, fatigue, dyspnea and depression at six to eight months after SARS-CoV-2 infection.** Panel (a) shows associations for not having fully recovered among initially symptomatic participants from multivariable logistic regression models adjusted for age, sex, initial hospitalization, symptom severity and presence of comorbidities. Panel (b) demonstrates associations for presence of fatigue (based on Fatigue Assessment Scale) from models adjusted for age, sex, and initial hospitalization. Panel (c) displays associations for presence of dyspnea (mMRC grade ≥1) from models adjusted for age, sex, initial hospitalization, smoking status, respiratory comorbidity and body mass index. Panel (d) shows associations for presence of depressive symptoms (based on DASS-21) from models adjusted for age, sex, initial hospitalization and symptom severity.

contrasts were observed for symptoms of anxiety. Meanwhile, younger participants and females more often reported stress symptoms compared to older individuals and males, respectively. In multivariable analyses, we only found lower education status and being unemployed to be associated with symptoms of depression (Fig 2 and S5 Table).

## Overlap of main outcomes and health-related quality of life

In total, 296 (69%) participants were categorized as non-recovered or experiencing fatigue, dyspnea or depression. Among these, 19 (6.4%) participants reported all four outcomes, while 130 (44%) suffered only from one of them (S1 Fig). Most frequent combinations were fatigue and depression (n = 94, 32%), fatigue and non-recovery (n = 78, 26%), and fatigue and mMRC grade ≥1 dyspnea (n = 68, 23%). Among all participants, 225 (53%) reported problems in at

least one EQ-5D-5L dimension. Most frequently affected dimensions were pain/discomfort (n = 151, 39%) and anxiety/depression (n = 133, 31%) (Table 2).

## Healthcare service utilization

A total of 170 (40%) participants reported having had at least one contact with the healthcare system for reasons related to COVID-19 (Table 3). Out of 81 initially hospitalized individuals, 8 (10%) were rehospitalized at least once due to persistent symptoms or COVID-19-related

**Table 3. Healthcare use and complications at six to eight months after SARS-CoV-2 infection.**

| Variable | Age group | | | Sex | | Hospitalization | | Overall, N = 431 |
|---|---|---|---|---|---|---|---|---|
| | 18–39 years, N = 164 | 40–64 years, N = 205 | ≥65 years, N = 62 | Female, N = 214 | Male, N = 217 | Non-hospitalized, N = 350 | Hospitalized, N = 81 | |
| **GP visit related to COVID-19** | 33 (20.4%) | 85 (42.7%) | 32 (53.3%) | 80 (38.5%) | 70 (32.9%) | 100 (29.2%) | 50 (63.3%) | 150 (35.6%) |
| *Missing* | *2* | *6* | *2* | *6* | *4* | *8* | *2* | *10* |
| **Number of GP visits related to COVID-19** [a] | | | | | | | | |
| 1–2 | 22 (67%) | 63 (75%) | 21 (66%) | 56 (71%) | 50 (71%) | 70 (71%) | 36 (72%) | 106 (71%) |
| 3–5 | 10 (30%) | 18 (21%) | 8 (25%) | 19 (24%) | 17 (24%) | 26 (26%) | 10 (20%) | 36 (24%) |
| ≥6 | 1 (3%) | 3 (4%) | 3 (9%) | 4 (5%) | 3 (4%) | 3 (3%) | 4 (8%) | 7 (5%) |
| *Missing* | *0* | *1* | *0* | *1* | *0* | *1* | *0* | *1* |
| **Medical hotline contact related to COVID-19** | 16 (9.8%) | 13 (6.3%) | 2 (3.3%) | 19 (8.9%) | 12 (5.6%) | 25 (7.2%) | 6 (7.4%) | 31 (7.2%) |
| *Missing* | *0* | *0* | *1* | *0* | *1* | *1* | *0* | *1* |
| **Number of medical hotline contacts related to COVID-19**[a] | | | | | | | | |
| 1–2 | 14 (88%) | 7 (70%) | 2 (100%) | 15 (88%) | 8 (73%) | 18 (78%) | 5 (100%) | 23 (82%) |
| 3–5 | 2 (12%) | 3 (30%) | 0 (0%) | 2 (12%) | 3 (27%) | 5 (22%) | 0 (0%) | 5 (18%) |
| ≥6 | 0 (0%) | 0 (0%) | 0 (0%) | 0 (0%) | 0 (0%) | 0 (0%) | 0 (0%) | 0 (0%) |
| *Missing* | *0* | *3* | *0* | *2* | *1* | *2* | *1* | *3* |
| **Rehospitalizations related to COVID-19** (N = 81) | 1 (10%) | 3 (7%) | 4 (14%) | 4 (11%) | 4 (10%) | - | 8 (10%) | 8 (10%) |
| **Number of rehospitalizations related to COVID-19** [a] | | | | | | | | |
| 1 | 0 (0%) | 3 (100%) | 4 (100%) | 3 (75%) | 4 (100%) | - | 7 (88%) | 7 (88%) |
| 2–3 | 1 (100%) | 0 (0%) | 0 (0%) | 1 (25%) | 0 (0%) | - | 1 (12%) | 1 (12%) |
| **GP visit or rehospitalization related to COVID-19** | 34 (21.0%) | 85 (42.7%) | 33 (55.0%) | 82 (39.4%) | 70 (32.9%) | 100 (29.2%) | 52 (65.8%) | 152 (36.1%) |
| *Missing* | *2* | *6* | *2* | *6* | *4* | *8* | *2* | *10* |
| **Healthcare contact related to COVID-19** | 45 (27.8%) | 92 (46.2%) | 33 (55.0%) | 94 (45.2%) | 76 (35.7%) | 116 (33.9%) | 54 (68.4%) | 170 (40.4%) |
| *Missing* | *2* | *6* | *2* | *6* | *4* | *8* | *2* | *10* |
| **New medical diagnoses** | 12 (7.3%) | 47 (22.9%) | 18 (29.0%) | 36 (16.8%) | 41 (18.9%) | 43 (12.3%) | 34 (42.0%) | 77 (17.9%) |
| **Type of new medical diagnosis** [b] | | | | | | | | |
| COVID-19 related complication (medically evaluated) | 2 (17%) | 17 (36%) | 8 (44%) | 12 (33%) | 15 (37%) | 12 (28%) | 15 (44%) | 27 (35%) |
| COVID-19 related complication (self-evaluated) | 3 (25%) | 6 (13%) | 2 (11%) | 6 (17%) | 5 (12%) | 7 (16%) | 4 (12%) | 11 (14%) |
| Non COVID-19 related diagnosis or unclear | 7 (58%) | 24 (51%) | 8 (44%) | 18 (50%) | 21 (51%) | 24 (56%) | 15 (44%) | 39 (51%) |

DASS-21 = Depression, Anxiety and Stress Score (21 items), EQ = EuroQol, FAS = Fatigue Assessment Scale, GP = General Practitioner, IQR = Interquartile Range, mMRC = modified Medical Research Council, VAS = Visual Analogue Scale.

[a] percentage within each category of healthcare contact (general practitioner visit, medical hotline call, rehospitalisation),

[b] percentage among all medical diagnoses.

complications. 224 (52%) participants reported at least one general practitioner visit for any reason, and 150 (36%) had a general practitioner visit related to COVID-19. Among those, the median number of COVID-19-related general practitioner visits was 2 (IQR 1 to 3). Older and initially hospitalized individuals more frequently reported having seen a general practitioner. Additionally, 31 (7%) participants reported to have called a medical hotline for a reason related to COVID-19. Among non-recovered participants, 33% (37/111) did not report any further healthcare contacts (S6 Table).

Since infection, 77 (18%) participants reported a new physician-diagnosed medical condition. 27 (35%) of these diagnoses were considered as related to COVID-19 by their physician (Table 3). Most frequently reported COVID-19-related conditions concerned the respiratory system (56%), followed by neuro-cognitive (30%), cardiovascular (11%), and skin disorders (11%). In multivariable analyses, we found evidence for an association between healthcare use and initial hospitalization, having experienced severe to very severe symptoms, female sex, and age ≥40 years (Fig 3 and S7 Table). Furthermore, not having fully recovered, grade ≥1

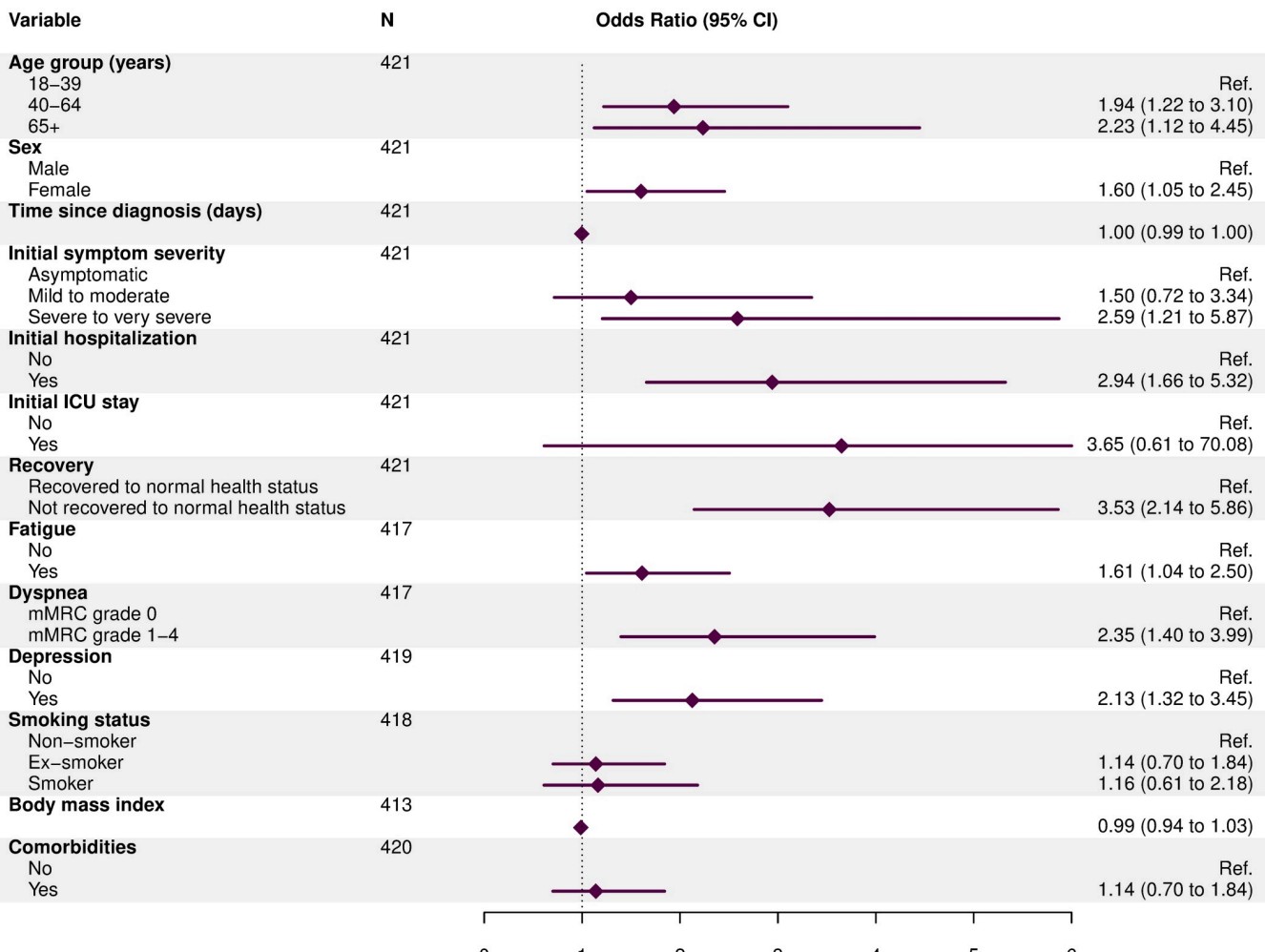

**Fig 3. Associations for healthcare service utilization at six to eight months after SARS-CoV-2 infection.** Fig 3 shows associations for having at least one further healthcare contact after initial COVID-19, based on multivariable logistic regression models adjusted for age, sex, initial hospitalization, and initial symptom severity.

dyspnea, fatigue and symptoms of depression were associated with further healthcare contacts after acute COVID-19.

## Sensitivity analyses

Sensitivity analyses of the main outcomes and health-related quality of life stratified into time periods of limited and increased testing, as well as limited and increased awareness of post-COVID-19 syndrome yielded similar results across the different time periods (S8 Table).

## Discussion

### Main findings

In this population-based cohort study, we found that one in four people had not fully recovered within six to eight months after SARS-CoV-2 infection. More than half of the participants in our study reported symptoms of fatigue. One fourth suffered from some degree of dyspnea or had symptoms of depression. Overall, more than two thirds had not recovered or experienced fatigue, dyspnea or depression at the time of follow-up, with only partial overlap between these outcomes. While not all of these outcomes are necessarily attributable to COVID-19, our study showed that an important proportion of infected individuals may develop post-COVID-19 syndrome and that a wide range of healthcare services may be required to support their needs.

Two fifths of study participants had at least one further healthcare contact related to COVID-19 after acute illness. 36% of participants reported further general practitioner visits, 7% calls to medical hotlines, and 10% of initially hospitalized participants were rehospitalized at least once for persistent symptoms or complications. Compared to recovered individuals, those not having fully recovered were more than three times more likely to have further healthcare contacts. These findings highlight the considerable long-term impact that COVID-19 may have both on affected individuals and healthcare systems worldwide.

### Evidence in context

The NICE guidelines defined post-COVID-19 syndrome as signs and symptoms developing during or after COVID-19 and continuing for more than 12 weeks [4]. Various studies have described a wide range of physical, cognitive and psychological symptoms persisting up to three months in individuals recovering from COVID-19 [24–33]. Yet, only few studies have assessed the persistence of symptoms beyond three months after infection [5–9].

Compared to studies that enrolled patients who were hospitalized for acute COVID-19 [5, 6], we observed a lower percentage of individuals suffering from longer-term symptoms. While these differences could be partly due to older participant populations and the restriction to hospitalized patients in their studies, we still observed lower proportions of non-recovery and persistent symptoms among hospitalized patients and older individuals. Meanwhile, a longitudinal cohort including 91% of participants with mild disease found persistent symptoms in 33% of outpatients and 31% of hospitalized patients [7]. These observations are more comparable to our findings. While differences in study populations and outcome measurement are likely to strongly affect the comparability of studies on post-COVID-19 syndrome, a relatively high prevalence of fatigue, dyspnea or exercise intolerance, and psychological symptoms have consistently been noted across studies [5–7, 12, 13].

Findings regarding longer-term sequalae are similar to those from prior coronavirus outbreaks [34], with 40% of severe acute respiratory syndrome (SARS) survivors reporting chronic fatigue up to four years after infection [35]. Similar chronic symptoms, in particular

fatigue, have been also described in other viral (e.g. Ebola virus, Epstein-Barr virus, Dengue virus), and bacterial (e.g. Borrelia burgdorferi) infections [36–40]. Results of two recent studies comparing outcomes in individuals with COVID-19 to individuals with influenza are suggestive of a higher burden of a wide range of longer-term sequalae associated with COVID-19 [8, 41].

Only few studies so far have described the utilization of healthcare services after COVID-19 [10, 11, 42–44]. At two months, 9% to 20% of patients hospitalized for COVID-19 were found to require rehospitalization, with a higher risk in older individuals and those with comorbidities [10, 11, 42–44]. The observed rehospitalization rate in our study is broadly consistent with these estimates. One study found that 78% of hospitalized participants had seen a primary care physician after hospital discharge for any reason after 2 months [11]. In our study, 63% of hospitalized participants reported having had a general practitioner visit after hospital discharge. As noted above, differences in study populations likely affect the comparison of results across studies. Further studies based on standardized assessments of health outcomes and symptom complexes will be necessary to capture the full spectrum of post-COVID-19 syndrome.

## Implications for healthcare resource planning

The management and care of individuals with post-COVID-19 syndrome is likely to become a substantial burden for healthcare systems worldwide. In Switzerland, 0.7 million individuals have been diagnosed with COVID-19 [45], and more than 2 million are estimated to have been infected according to current seroprevalence studies [46]. Based on our estimates, a relevant number of individuals suffering from longer-term complications has to be expected, which will require some degree of support or healthcare services. In our study, we provide more detailed data on healthcare utilization incurring due to COVID-19. More than a third of infected individuals in our study needed an average of two further primary care consultations related to protracted symptoms or complications. Interestingly, we also observed that despite an increased likelihood of seeking care in those who have not returned to their normal health status, approximately one third of these participants did not report any further healthcare contact after their acute illness. This indicates that there may be a relevant need among previously infected individuals for additional services specialized in the care of people with post-COVID-19 syndrome. Our study provides important evidence for understanding the longer-term complications and burden of COVID-19 on healthcare systems and for planning public health resources and tailored services accordingly.

## Limitations

By relying on official records of all diagnosed infections, our study provided a unique opportunity to evaluate post-COVID-19 syndrome in the general population, based on the full spectrum of disease severity. However, our study also has several limitations.

First, most participants included in this analysis were diagnosed with COVID-19 during the first pandemic wave in Switzerland. The capacity constraints in SARS-CoV-2-testing up to June 2020 may have selected for a population with a higher risk of experiencing severe disease as only those qualified for testing at that time. Furthermore, increased awareness of post-COVID-19 syndrome may have resulted in more frequent reporting of health issues by participants. However, sensitivity analyses stratified by time periods of limited and increased testing and limited and increased awareness of post-COVID-19 syndrome did not show a relevant difference between the respective time periods (S8 Table).

Second, self-selection bias may have occurred if individuals who are more concerned with their health or experiencing symptoms related to post-COVID-19 syndrome were more likely to participate. This may have biased our results towards higher estimates of non-recovery and healthcare use than in the full population of infected individuals. Furthermore, the primarily electronic setup of our study may have influenced participation. On one side, this may have led to an underrepresentation of older individuals and those with difficulties using the electronic platform, as well as those with severe impairments due to post-COVID-19 syndrome or other conditions. We undertook strong efforts to include such individuals by establishing repeated contacts via phone prior to enrolment and encouraging the support by relatives for electronic surveys and alternatively offering phone interviews. In our study, the proportion of older individuals and individuals initially hospitalized for COVID-19 was lower in our study compared to nonparticipants (S1 Table). This may have biased the results towards lower estimates of non-recovery and healthcare use. On the other side, the digital nature of the follow-up may have facilitated the recruitment of participants who prefer not to present for in-person study visits (e.g., for convenience or health and mobility issues) [47]. Compared to studies relying on study site visits, the electronic setup may thus have increased the diversity of the study population. Overall, it is thus difficult to estimate the magnitude and direction of potential biases arising from participant selection. Nevertheless, we also consider the population-based approach a strength of our study.

Third, we did not have a baseline (pre-COVID-19) assessment of participants' physical and mental health. Thus, it is impossible to distinguish the effects of COVID-19 from pre-existing conditions. The interpretation of our findings regarding depression and anxiety is further limited by the psychological burden that the pandemic may impose in general [48, 49]. While we tried to compare our results with estimates from the general population, applicable comparison data was not available. Other studies investigating longer-term sequelae after SARS-CoV-2 infection found a relevant excess risk for longer-term symptoms among infected individuals compared to SARS-CoV-2-negative control groups [9, 50]. Further research is required to gain better insights into the disease and healthcare burden attributable to SARS-CoV-2 infection.

Last, we did not evaluate the use of specialized medical (e.g., psychological/psychiatric care) or diagnostic services in our assessment. Thus, the true extent of healthcare service utilization may be underestimated. The unavailability of targeted post-COVID-19 care programs in Switzerland at the time of enrolment may have led to an underestimation of the healthcare demand to be expected once such programs become available. Additionally, it is important to consider case detection rates and population subgroups infected when estimating the impact of COVID-19 on healthcare systems. In contexts with limited testing and detection of infected individuals, the need for specialized healthcare services may be underestimated without adjustment for underdetection. Furthermore, the spread of SARS-CoV-2 likely varied across countries regarding which population groups were primarily affected, which may also influence the expected burden of post-COVID-19 syndrome on healthcare systems in other contexts.

## Conclusion

Our population-based cohort study showed that a considerable proportion of SARS-CoV-2 infected individuals experience longer-term consequences and have a relevant demand for healthcare services. A wide range of services and patient-centered, integrative approaches will be required to support the recovery of these individuals. It is thus crucial to timely allocate resources and plan healthcare services to respond to the needs of those suffering from post-COVID-19 syndrome.

## Supporting information

**S1 Table. Comparison of population characteristics of participants of the Zurich SARS-CoV-2 Cohort study and individuals not participating in the study.**
(DOCX)

**S2 Table. Results from univariable and multivariable logistic regression models for the outcome of not having fully recovered at six to eight months after diagnosis.**
(DOCX)

**S3 Table. Results from univariable and multivariable logistic regression models for the outcome of fatigue at six to eight months after diagnosis.**
(DOCX)

**S4 Table. Results from univariable and multivariable logistic regression models for the outcome of mMRC dyspnea grade ≥1 at six to eight months after diagnosis.**
(DOCX)

**S5 Table. Results from univariable and multivariable logistic regression models for the outcome of depression at six to eight months after diagnosis.**
(DOCX)

**S6 Table. Overlap of participants not having recovered or experiencing fatigue, dyspnea, or depression and healthcare use at six to eight months after diagnosis.**
(DOCX)

**S7 Table. Results from univariable and multivariable logistic regression models for the outcome of having at least one further healthcare contact (defined as rehospitalization, general practitioner visit or medical hotline call) related to COVID-19 within six to eight months after diagnosis.**
(DOCX)

**S8 Table. Sensitivity analysis of relative health status, fatigue, dyspnea, mental health, and health-related quality of life in study participants at six to eight months after SARS-CoV-2 infection, stratified into time periods of limited and increased testing for SARS-CoV-2, as well as with limited and increased awareness of post-COVID-19 syndrome.**
(DOCX)

**S1 Fig. Venn diagram of the overlap of participants that have not fully recovered or are experiencing fatigue, dyspnea or depression at six to eight months after SARS-CoV-2 infection (total N = 296).**
(DOCX)

**S1 File. Original de-identified dataset underlying the analyses included in the study.**
(XLSX)

## Acknowledgments

We would like to thank the study administration staff and the staff of the Corona Center of the University of Zurich for their excellent work and dedication to this study. Furthermore, we thank the Department of Health of the Canton of Zurich for their support and collaboration in realizing the study. And last, we thank all the study participants for their valuable time and commitment to the Zurich SARS-CoV-2 Cohort.

## Author Contributions

**Conceptualization:** Dominik Menges, Tala Ballouz, Alexia Anagnostopoulos, Hélène E. Aschmann, Jan S. Fehr, Milo A. Puhan.

**Data curation:** Dominik Menges, Tala Ballouz.

**Formal analysis:** Dominik Menges, Tala Ballouz.

**Funding acquisition:** Jan S. Fehr, Milo A. Puhan.

**Investigation:** Dominik Menges, Tala Ballouz, Alexia Anagnostopoulos, Hélène E. Aschmann, Anja Domenghino, Milo A. Puhan.

**Methodology:** Dominik Menges, Tala Ballouz, Hélène E. Aschmann, Jan S. Fehr, Milo A. Puhan.

**Project administration:** Dominik Menges, Tala Ballouz, Hélène E. Aschmann, Anja Domenghino, Milo A. Puhan.

**Supervision:** Jan S. Fehr, Milo A. Puhan.

**Validation:** Dominik Menges, Tala Ballouz, Milo A. Puhan.

**Visualization:** Dominik Menges, Tala Ballouz.

**Writing – original draft:** Dominik Menges, Tala Ballouz, Milo A. Puhan.

**Writing – review & editing:** Dominik Menges, Tala Ballouz, Alexia Anagnostopoulos, Hélène E. Aschmann, Anja Domenghino, Jan S. Fehr, Milo A. Puhan.

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
