## [Decision Letter · Decision Letter 0]

13 May 2021

PONE-D-21-12057

Burden of Post-COVID-19 Syndrome and Implications for Healthcare Service Planning: A Population-based Cohort Study

PLOS ONE

Dear Dr. Puhan,

Thank you for submitting your manuscript to PLOS ONE. After careful consideration, we feel that it has merit but does not fully meet PLOS ONE’s publication criteria as it currently stands. Therefore, we invite you to submit a revised version of the manuscript that addresses the points raised during the review process.

Based on the comments from the reviewers, I am recommending that you make substantial revisions to your manuscript. You will need to pay more attention to your study designs and adequately describe the inclusion and exclusion criteria for patient recruitment. It is evident in this manuscript that the results may be biased because the way the patients were recruited. Please include this issue in your discussion by pointing out the limitations of the presented results. You also carried out a sensitivity analysis but there is no mention of this in the results. Please submit your revised manuscript after you have attended to all the reviewer's comments as advised in this letter.

We look forward to receiving your revised manuscript.

Kind regards,

Martin Chtolongo Simuunza, PhD

Academic Editor

PLOS ONE

Journal Requirements:

Reviewers' comments:

Reviewer's Responses to Questions

**Comments to the Author**

1. Is the manuscript technically sound, and do the data support the conclusions?

Reviewer #1: Yes

Reviewer #2: Yes

Reviewer #3: Yes

2. Has the statistical analysis been performed appropriately and rigorously? 

Reviewer #1: I Don't Know

Reviewer #2: Yes

Reviewer #3: Yes

3. Have the authors made all data underlying the findings in their manuscript fully available?

Reviewer #1: Yes

Reviewer #2: Yes

Reviewer #3: Yes

4. Is the manuscript presented in an intelligible fashion and written in standard English?

Reviewer #1: Yes

Reviewer #2: Yes

Reviewer #3: Yes

5. Review Comments to the Author

Reviewer #1: This study addresses a relevant issue: long-term sequelae of COVID-19.

Of 4639 SARS-CoV-2 positive patients, 2209 could be followed up and 1309 fulfilled the inclusion criteria.

Of these, 442 consented to the studies and 431 were finally included.

Of these, 385 reported continued symptoms or impairment due to the previous COVID-19.

Nevertheless, the study leaves some questions in my opinion:

The very high rate of patients reporting symptoms (385/431) suggests that there is a significant bias.

Can anything be said from the health data about the ≈900 patients who met the inclusion criteria but did not participate (anonymized analysis of age, sex, hospitalization rate during acute illness, comorbidities, social class, etc)?

In the absence of baseline data (the authors correctly note this), it is difficult to quantify a "true" effect of COVID-19.

The excellent health care system in Switzerland may lead to a particularly fine-grained resolution of the data compared to the rest of the world. The number of patients evaluated is very small compared to the total number of patients affected in Switzerland, Europe or worldwide.

A comparison to patients with similar diseases (e.g. influenza, pneumococcal pneumonia) would be helpful to compare the results. From studies in the setting pulmonary, medicine, infectious diseases or intensive care medicine, it is known that the rate of patients who still report symptoms and limitations months after the acute illness is high.

Reviewer #2: Comments

Overall Comment

Very good study, informative and timely. Only minor clarifications and corrections requested.

Comment 1. Line 229, Kindly clarify the direction of the association with Body Mass Index.

Comment 2. There is no comment on sensitivity analysis in the results or discussion.

Comment 3. Kindly clarify. In the third condition of sensitivity analysis, the period with high public awareness of post-COVID-19 syndrome (period after 09 November 2020) leaves no participants positive in that period to be selected since you only considered individuals who tested positive between 27th February 2020 and 5th August 2020.

Reviewer #3: It is an interesting work, which analyzes the clinical evolution of patients with COVID in the long term (7 months)

It presents fundamental biases in the selection of the population, which make the results difficult to extrapolate to other populations

In the first place, the selection is made with patients who are able to fill out the electronic form, which may, on the one hand, limit access to older people or people with more associated pathology.

Second, a third of the patients that have been selected are recruited. It is not described whether the characteristics (at least age, sex, hospitalization) of this population are similar to the population that was included in the study.

Likewise, this population willing to participate in the survey may overestimate persistent symptoms, since those patients with symptoms will tend to answer more frequently

In this way, the population included is young, although a third with comorbidity, and hospital admission in a very significant proportion

What is interesting about the study is the stratification of symptoms, and the evaluation of the overlap of dyspnea, fatigue, or non-recovery with depressive symptoms

It is striking that many of the symptoms are associated with initial severity but not with hospitalization.

The description of the series, with stratification of long-term symptoms, evaluation of the demand for health care and analysis of associated factors, offers relevant information

But the selection bias of this population should be pointed out more intensively in the limitations.

Likewise, I do not believe that the data allow us to offer a real vision of healthcare needs. These can only be estimated, if a population is evaluated without so many selection biases, so a reflection on this message should be made

The message cannot be transferred to the global population of COVID patients, given the previously mentioned biases, so it should be qualified in the Implications for healthcare resource planning section

6. PLOS authors have the option to publish the peer review history of their article (what does this mean?). If published, this will include your full peer review and any attached files.

Reviewer #1: No

Reviewer #2: No

Reviewer #3: No

---

## [Author Response · Author response to Decision Letter 0]

9 Jun 2021

Editor comments

Based on the comments from the reviewers, I am recommending that you make substantial revisions to your manuscript. You will need to pay more attention to your study designs and adequately describe the inclusion and exclusion criteria for patient recruitment. It is evident in this manuscript that the results may be biased because the way the patients were recruited. Please include this issue in your discussion by pointing out the limitations of the presented results. You also carried out a sensitivity analysis but there is no mention of this in the results. Please submit your revised manuscript after you have attended to all the reviewer's comments as advised in this letter.

Many thanks for your comments and suggestions. We agree with you and the reviewers that potential selection bias arising from the recruitment of our participants may have important implications for the interpretation of our findings. We have further expanded on this limitation in the discussion (see lines 392-410). Regarding the results from sensitivity analysis, we had briefly discussed these in lines 388-390 (limitations section). We now additionally report on sensitivity analyses in the results section (lines 297-300).

Additional Journal Requirements

The manuscript has been edited to meet the style requirements of PLOS ONE. Please find the changes highlighted within the text. As part of this revision, the supplementary material has been reordered and is now provided in individual files.

Additional details regarding informed consent have been added to the manuscript (line 98) and in the submission form. 

We will share de-identified individual participant data as part of this resubmission (Supplementary File S10). Please note that we cannot provide the exact participant age and the stratification variables used in sensitivity analysis, since these could allow the triangulation and indirect identification of individuals included in our study during time periods with low SARS-CoV-2 case numbers. All presented main analyses can be reproduced with the data provided.

The ethics section is now included only in the Methods section.

Captions for Supporting Information have been added at the end of the manuscript.

Reviewers' comments

Reviewer #1:

This study addresses a relevant issue: long-term sequelae of COVID-19. Of 4639 SARS-CoV-2 positive patients, 2209 could be followed up and 1309 fulfilled the inclusion criteria.

Of these, 442 consented to the studies and 431 were finally included.

Of these, 385 reported continued symptoms or impairment due to the previous COVID-19.

Nevertheless, the study leaves some questions in my opinion:

The very high rate of patients reporting symptoms (385/431) suggests that there is a significant bias.

Can anything be said from the health data about the ≈900 patients who met the inclusion criteria but did not participate (anonymized analysis of age, sex, hospitalization rate during acute illness, comorbidities, social class, etc)?

Many thanks for this valuable feedback. We agree that potential sources for selection bias are an important consideration in interpreting the findings of our study. As was previously reported in the discussion section (moved now to the results section, lines 195-197) and in S9 Table (which provides a comparison of the enrolled and non-enrolled participants), our population was younger and a lower proportion was hospitalized secondary to COVID-19 (19% versus 24%). Additionally, a higher proportion of our study participants had symptoms (89.3% compared to 80.7% in nonparticipants). However, it is to be noted that symptom assessment in non-participants is based on data collected by contact tracers as well as mandatory reporting by physicians and laboratories before contact tracing was fully established in May 2020. Due to these circumstances, data from individuals infected before May 2020 is incomplete. Additionally, we have no information on whether the non-participants developed symptoms after the initial contact tracing call, which may partly explain the lower proportion of symptomatic non-participants. While further information on the socio-demographic characteristics of non-participants would have been very useful, such data was not collected/available. 

In the absence of baseline data (the authors correctly note this), it is difficult to quantify a "true" effect of COVID-19.

The excellent health care system in Switzerland may lead to a particularly fine-grained resolution of the data compared to the rest of the world. The number of patients evaluated is very small compared to the total number of patients affected in Switzerland, Europe or worldwide.

A comparison to patients with similar diseases (e.g. influenza, pneumococcal pneumonia) would be helpful to compare the results. From studies in the setting pulmonary, medicine, infectious diseases or intensive care medicine, it is known that the rate of patients who still report symptoms and limitations months after the acute illness is high.

This is equally an important point. From the data collected in our study, it is not possible to estimate the "true effect". As stated in the manuscript, we attempted to use reference health data from Switzerland to estimate the proportion of individuals with symptoms attributable to SARS-CoV-2 infection. However, no applicable health data was available. Certainly, a comparison with other diseases could provide additional context, and we agree that other diseases may have similar longer-term effects on physical and mental health. We included a brief discussion on this subject in the manuscript (lines 341-347). However, we consider a detailed discussion to be unlikely to improve the estimation of the "true effect" of SARS-CoV-2 infection due to the same issues with transportability and representativeness of study populations. Thus, we do not believe that the comparison with other illnesses would influence the message of our manuscript targeted at the public health response to COVID-19 specifically.

Reviewer #2:

Overall Comment

Very good study, informative and timely. Only minor clarifications and corrections requested.

Comment 1. Line 229, Kindly clarify the direction of the association with Body Mass Index.

Many thanks for your valuable feedback on our manuscript. Grade ≥1 dyspnea was associated with higher body mass index- we have added the direction of the association to the text (line 388). 

Comment 2. There is no comment on sensitivity analysis in the results or discussion.

Thank you for pointing this out. Please note that we had commented on the sensitivity analysis in the limitation section within the discussion (lines 388-390). However, we now also briefly report on this analysis in the results section (see lines 297-300). 

Comment 3. Kindly clarify. In the third condition of sensitivity analysis, the period with high public awareness of post-COVID-19 syndrome (period after 09 November 2020) leaves no participants positive in that period to be selected since you only considered individuals who tested positive between 27th February 2020 and 5th August 2020.

Many thanks for this comment. The time period of “awareness of post-COVID-19 syndrome” that we refer to is not related to the time of SARS-CoV-2 diagnosis but rather the time of recruitment into the study and completion of the questionnaire (5.9 to 10.3 months after diagnosis corresponds to October 2020 to January 2021). To avoid any confusion for the readers, we have now made this explicitly clear in the text (line 167). 

Reviewer #3: 

It is an interesting work, which analyzes the clinical evolution of patients with COVID in the long term (7 months)

It presents fundamental biases in the selection of the population, which make the results difficult to extrapolate to other populations

In the first place, the selection is made with patients who are able to fill out the electronic form, which may, on the one hand, limit access to older people or people with more associated pathology.

Second, a third of the patients that have been selected are recruited. It is not described whether the characteristics (at least age, sex, hospitalization) of this population are similar to the population that was included in the study.

Likewise, this population willing to participate in the survey may overestimate persistent symptoms, since those patients with symptoms will tend to answer more frequently

In this way, the population included is young, although a third with comorbidity, and hospital admission in a very significant proportion.

Many thanks for your valuable feedback on our manuscript and this important comment. We agree that potential sources for selection bias are an important consideration in interpreting the findings of our study. The primarily electronic nature of our study may well have influenced participation in our study. In the results section (lines 195-197) and in S9 Table (which provides a comparison of the enrolled and non-enrolled participants), we provided a comparison of participants and non-participants. In brief, our population was younger and a lower proportion was hospitalized secondary to COVID-19 (19% versus 24%). Thus, your concerns are certainly valid. However, this study also constitutes one of very few in which sampling of participants was based on a population-based sample among all identified cases, while many others were based exclusively on hospitalized (or hospital-diagnosed) or purposive samples suffering from similar issues. As such, our enrollment process is well documented and transparent with respect to these limitations. Furthermore, the digital nature of the follow-up may also have facilitated the recruitment of participants who would prefer not to present for in-person study visits (e.g., for convenience or health and mobility issues). For example, experiences from the Multiple Sclerosis registry in Switzerland have shown that digital questionnaires may even increase the diversity of the study population (e.g. those with severe disease; https://doi.org/10.4414/smw.2018.14623). We did everything possible to enable everyone to participate, e.g. by having repeated phone contacts prior to the use of the electronic enrollment system (encouraging help by family members) and performing phone interviews with individuals that were unable to use a computer. Our experience was that participation and motivation among those that we were able to contact was exceptionally high compared to other studies we have conducted due to the high public interest, regardless of the personal burden. However, we have revised the limitations section to make these considerations more explicit (lines 392-410).

What is interesting about the study is the stratification of symptoms, and the evaluation of the overlap of dyspnea, fatigue, or non-recovery with depressive symptoms

It is striking that many of the symptoms are associated with initial severity but not with hospitalization.

The description of the series, with stratification of long-term symptoms, evaluation of the demand for health care and analysis of associated factors, offers relevant information

But the selection bias of this population should be pointed out more intensively in the limitations.

Likewise, I do not believe that the data allow us to offer a real vision of healthcare needs. These can only be estimated, if a population is evaluated without so many selection biases, so a reflection on this message should be made

The considerations about selection bias certainly also apply to healthcare needs. However, it is difficult to estimate the direction of any potential bias. On one side, older and more severely ill individuals may be underrepresented, leading to estimates that are too low. On the other side, individuals with issues may be more likely to participate, leading to estimates that are too high. We expanded on this issue in the limitations section (lines 426-432). 

The message cannot be transferred to the global population of COVID patients, given the previously mentioned biases, so it should be qualified in the Implications for healthcare resource planning section

Please see above - we extended on this in the limitations section. Our study contributes to the evidence on post-COVID-19 syndrome through data from a unique, population-based sample. More research is needed to summarize all available evidence and provide meta-estimates to estimate the "true" effects of the pandemic. However, in light of current systematic reviews, our estimates seem to lie more or less in the middle of what has already been published. We are positive that this work adds important evidence to the current literature.

---

## [Decision Letter · Decision Letter 1]

18 Jun 2021

PONE-D-21-12057R1

Burden of post-COVID-19 syndrome and implications for healthcare service planning: A population-based cohort study

PLOS ONE

Dear Dr. Puhan,

Thank you for submitting your manuscript to PLOS ONE. After careful consideration, we feel that it has merit but does not fully meet PLOS ONE’s publication criteria as it currently stands. Therefore, we invite you to submit a revised version of the manuscript that addresses the points raised during the review process.

The reviewers are of the view that you have made substantial revisions that has greatly improved the manuscript. However, there are a few other minor concerns that you will have to attend to that they think will further improve the manuscript. I consider all the suggestions to be very important and should be attended to. However of major concern to me is the choice of the reference category especially for nominal variables, as it will also improve the precision of the estimates. Please attend to them and resubmit your manuscript as advised in this letter.

We look forward to receiving your revised manuscript.

Kind regards,

Martin Chtolongo Simuunza, PhD

Academic Editor

PLOS ONE

Journal Requirements:

Reviewers' comments:

Reviewer's Responses to Questions

**Comments to the Author**

1. If the authors have adequately addressed your comments raised in a previous round of review and you feel that this manuscript is now acceptable for publication, you may indicate that here to bypass the “Comments to the Author” section, enter your conflict of interest statement in the “Confidential to Editor” section, and submit your "Accept" recommendation.

Reviewer #1: All comments have been addressed

Reviewer #2: (No Response)

2. Is the manuscript technically sound, and do the data support the conclusions?

Reviewer #1: Yes

Reviewer #2: Yes

3. Has the statistical analysis been performed appropriately and rigorously? 

Reviewer #1: I Don't Know

Reviewer #2: Yes

4. Have the authors made all data underlying the findings in their manuscript fully available?

Reviewer #1: Yes

Reviewer #2: Yes

5. Is the manuscript presented in an intelligible fashion and written in standard English?

Reviewer #1: Yes

Reviewer #2: Yes

6. Review Comments to the Author

Reviewer #1: After the revision the manuscript „Burden of post-COVID-19 syndrome and implications for healthcare service planning: a population-based cohort study” (PONE-D-21-12057_R1) by Menges and colleagues has improved substantially. My questions were answered satisfactorily for the most part. The remaining gaps are adequately discussed by the authors in the limitations.

Please correct line 417 “On one side,...” instead of “One one side,…”

Reviewer #2: All my previous comments were answered and I am satisfied with the responses. I just have the following final comments:

Comment 1: Line 168 – 170 the sentence looks incomplete. I think it is missing the word “bias”. As in “to assess potential selection bias”.

Comment 2: Table 1 Kindly put units in brackets for the variable “initial symptom duration” (days). just like you had put for the variable “time since diagnosis (days”.

Comment 3: The multivariate logistics regression analysis results in table 2 are ok and well presented. However, there is possible misallocation of the reference group or level for some of the significant covariables. For example, the covariable “gender” for outcome “non-recovery” and “dyspnea”. Also the covariable age group for outcome “fatigue” etc. It is preferable that the level with the lower proportion of the outcome variable should be the reference level for that covariable. However, this does not change the significance of the results and it may only be a matter of preference of reporting.

Comment 4: looking at the results in figure 2, isn’t the covariable “presence of comorbidity” also significantly associated with outcome “dyspnea”?

7. PLOS authors have the option to publish the peer review history of their article (what does this mean?). If published, this will include your full peer review and any attached files.

Reviewer #1: No

Reviewer #2: No

---

## [Author Response · Author response to Decision Letter 1]

21 Jun 2021

Journal Requirements

-> We revised the publication list and updated the references for Sudre et al., 2021 (replaced medrxiv preprint with publication in Nature Medicine), as well as Mandal et al., 2021 and Al-Aly et al., 2021 (updated reference to most recent record). To our best knowledge, none of the cited articles has been retracted.

Reviewer #1

After the revision the manuscript „Burden of post-COVID-19 syndrome and implications for healthcare service planning: a population-based cohort study” (PONE-D-21-12057_R1) by Menges and colleagues has improved substantially. My questions were answered satisfactorily for the most part. The remaining gaps are adequately discussed by the authors in the limitations.

-> Thank you very much for your thorough review of our study and your positive feedback. We are happy that we were able to address your concerns to your satisfaction.

Please correct line 417 “On one side,...” instead of “One one side,…”

-> We have corrected this mistake.

Reviewer #2

All my previous comments were answered and I am satisfied with the responses. I just have the following final comments:

-> Thank you very much for your insightful comments and your positive feedback. We are happy that we were able to satisfactorily respond to your concerns.

Comment 1: Line 168 – 170 the sentence looks incomplete. I think it is missing the word “bias”. As in “to assess potential selection bias”.

-> We changed the wording to "to assess potential selection bias.".

Comment 2: Table 1 Kindly put units in brackets for the variable “initial symptom duration” (days). just like you had put for the variable “time since diagnosis (days”.

-> Thank you for raising this to our attention. We included the units for initial symptom duration (days) and BMI (kg/m2) in Table 1, as well as for time from symptom onset to diagnosis (days) in Table S1.

Comment 3: The multivariate logistics regression analysis results in table 2 are ok and well presented. However, there is possible misallocation of the reference group or level for some of the significant covariables. For example, the covariable “gender” for outcome “non-recovery” and “dyspnea”. Also the covariable age group for outcome “fatigue” etc. It is preferable that the level with the lower proportion of the outcome variable should be the reference level for that covariable. However, this does not change the significance of the results and it may only be a matter of preference of reporting.

-> We agree that the results for regression analyses may be easier to interpret with a different allocation of the reference group for some variables. We changed the reference for sex in all analyses (Fig 2-3; S2-S5 Table, S8 Table), in line with your suggestions and the reporting in the text. Meanwhile, our preference would be to keep the current reporting for the other variables, for the following reasons: First, we would favor to use consistent reference levels for variables with varying direction of associations across analyses for the different outcomes (e.g. for age group, ICU stay). Second, we would prefer using the lowest levels for variables of ordinal nature (e.g. for age group, income, education), and the most frequent level (e.g. "employed") for employment. We investigated your suggestion and feel that our data is more interpretable as it is currently presented. Also, the precision of the respective estimates is negatively affected when changing the reference level (e.g. since the group of students and university-educated individuals is rather small). In our resubmission, we include both our suggested version for the figures and tables, as well as an alternative version (version 2) more in line with your suggestions. We would like to leave it up to the editor to decide which presentation of the data is preferred for publication.

Comment 4: looking at the results in figure 2, isn’t the covariable “presence of comorbidity” also significantly associated with outcome “dyspnea”?

-> This is correct - we added a statement regarding associations of the presence of comorbidities and chronic respiratory conditions with dyspnea to the manuscript (lines 246-248).

---

## [Editor Report · Decision Letter 2]

29 Jun 2021

Burden of post-COVID-19 syndrome and implications for healthcare service planning: A population-based cohort study

PONE-D-21-12057R2

Dear Dr. Puhan,

We’re pleased to inform you that your manuscript has been judged scientifically suitable for publication and will be formally accepted for publication once it meets all outstanding technical requirements.

Kind regards,

Martin Chtolongo Simuunza, PhD

Academic Editor

PLOS ONE
---

## [Editor Report · Acceptance letter]

2 Jul 2021

PONE-D-21-12057R2 

Burden of post-COVID-19 syndrome and implications for healthcare service planning: A population-based cohort study 

Dear Dr. Puhan:

I'm pleased to inform you that your manuscript has been deemed suitable for publication in PLOS ONE. Congratulations! Your manuscript is now with our production department. 

Kind regards, 

on behalf of

Dr. Martin Chtolongo Simuunza 

Academic Editor

PLOS ONE